# Detection and Segmentation of Solar Farms in Satellite Imagery: A Study of Deep Neural Network Architectures

## Abstract

In line with global sustainability goals, such as the Paris Agreement, accurate mapping and monitoring of solar farms are critical for achieving net zero emissions by 2050. However, many solar installations remain undocumented, posing a challenge. This work introduces Solis-seg, a Deep Neural Network optimized for detecting solar farms in satellite imagery. Solis-seg achieves a mean Intersection over Union (IoU) of 96.26% on a European dataset, outperforming existing solutions.

The study leans heavily on advances in semantic segmentation and NAS for solar farm detection. Semantic segmentation has evolved through technologies like Fully Convolutional Network (FCN) and U-Net, which have shown strong performance on satellite imagery. In NAS, Differentiable Architecture Search (DARTS) and its variants like Auto-DeepLab (ADL) have become efficient ways to automate the creation of architectures. This study also challenges the prevailing method of using transfer learning from classification tasks for semantic segmentation, suggesting new avenues for research.

Thus, this work contributes to both the field of earth observation machine learning and the global transition to renewable energy by providing an efficient, scalable solution for tracking solar installations. We believe that our research offers valuable insights into the application of advanced machine learning techniques for solar farm detection and also encourages further exploration in earth observation and sustainability.

## 1 Introduction

**Context.** With the Paris Agreement of 2015, a vast majority of nations globally have committed to reaching net zero emissions by 2050. Achieving this monumental goal requires a large-scale transition from fossil fuels towards renewable energy alternatives like solar and wind power. Currently, fossil fuels are responsible for nearly 80% of the global energy consumption and emit over 14 gigatonnes of $CO_2$, as reported by the International Energy Agency IEA (2022). The shift towards green energy sources such as wind, hydro, and solar is fundamental to meeting the Paris Agreement's climate objectives within the prescribed timeline. Non-compliance with these objectives could lead to catastrophic impacts on human civilization. In this work, we focus on solar energy production and in particular the problem of solar farm identification from satellite imagery.

**Challenges.** Despite the demonstrated prowess of Neural Architecture Search (NAS) in surpassing human-designed architectures in image classification competition datasets Elsken et al. (2018), its application in the field of solar farm identification from satellite imagery remains uncharted territory. Furthermore, while NAS has seen extensive use in well-established benchmarks, its practical application for novel datasets is still under-researched White et al. (2023). Recognizing these challenges, our study embarks on a multifaceted mission and makes several contributions.

**Contributions.** Building upon previous research Costa et al. (2021); Layman (2019), which investigates different architecture performances but does not explore NAS-derived solutions, we study NAS optimization for the real-world semantic segmentation task of solar farms and assess its broader performance beyond established benchmarks. In the process, we critically re-evaluate the strategy

of using transfer learning to develop segmentation networks from classification models with the purpose of locating solar farms in satellite imagery Yu et al. (2018); Kruitwagen et al. (2021); Hou et al. (2019). Our lessons learned may have implications for both solar farm segmentation and the wider application of NAS.

## 2 BACKGROUND

**Identifying Solar Farms from Images.** Several studies have explored the detection of solar panels on satellite imagery, utilizing both ANNs and other methods. For instance, a random forest model was employed by Plakman et al. Plakman et al. (2022) to detect solar panels, and this model was trained and evaluated using a publicly accessible dataset from the Netherlands. Hou et al. developed SolarNet, a system that integrates the merits of Expectation-Maximization Attention Networks and a U-Net architecture, to uncover new photovoltaic (PV) systems in China Hou et al. (2019). Meanwhile, in Brazil, a study used high-performing segmentation models with different pre-trained backbones Costa et al. (2021).

A group from Stanford has identified and compiled large-scale solar platforms and rooftop solar installations in the US into the publicly accessible DeepSolar database Yu et al. (2018)[1]. Astraea Earth trained a Deep Convolutional Neural Network in the US and utilized it to identify new solar farms in China Layman (2019).

One particularly significant contribution is the paper by Kruitwagen et al. (2021). Along with the paper, they released a global dataset of solar energy facilities, which expanded the existing asset-level data by an impressive 432%. This work represents the most substantial single contribution to this field to date, measured by the number of previously unknown facilities discovered and added to public datasets. Focusing on PV platforms larger than $10\,000\ m^2$, they achieved a precision of 98.6%, a recall of 90%, and an Intersection over Union (IoU) of 90% for the segmentation task on their test set. They employed a U-Net-based CNN model and used two sources of remote sensing imagery to achieve these results. Importantly, they leveraged the non-visible bands of Sentinel-2, demonstrating their significant role in the model's solar panel recognition.

**Semantic Segmentation.** Semantic segmentation is an area where CNNs have exhibited substantial success, highlighted by the victory of the Fully Convolutional Network (FCN) Long et al. (2014) in the COCO segmentation task in 2014. This achievement was credited to replacing the fully connected layers at the end of popular networks like AlexNet, VGG, and GoogLeNet with convolutional layers. This modification led to significant speed increases during both forward and backward passes in training Long et al. (2014). The method employs upsampling techniques to restore the output feature map of the image to its original size for pixel-by-pixel predictions.

U-Net further improved in 2017 by incorporating the output before each subsampling stage as input during the upsampling phase. This enhancement aids in more accurately mapping recognized features back to the original image size Ronneberger et al. (2015). As per Tao et al. (2022), U-Net is particularly effective for semantic segmentation on remote sensing imagery due to its superior performance with less training data in comparison to other algorithms. This can be an advantage if the original dataset is very small. Hou et al. (2019) and Kruitwagen et al. (2021) both use a U-Net for semantic segmentation of solar farms.

Dilated convolutions, also referred to as "atrous" convolutions, are a variant of convolutional neural network (CNN) layers that utilize dilated kernels to enlarge the receptive field of a layer without augmenting the number of parameters Chen et al. (2017). Traditional CNNs determine the receptive field of a layer based on its filter size and stride. However, dilated convolutions employ filters with gaps or "dilations," the size of which is decided by the dilation rate, enabling the filters to cover a larger input area without augmenting the number of parameters or computational complexity. This characteristic is particularly beneficial for semantic segmentation, where maintaining spatial resolution while increasing receptive field to capture long-range dependencies in data is crucial Garcia-Garcia et al. (2017).

**Neural Architecture Search.** The roots of Neural Architecture Search (NAS) can be traced back to 1989, when an evolutionary algorithm was first applied by Miller et al. (1989) to optimize neural

---

[1] https://deepsolar.web.app/

network architectures. Since that seminal work, an array of diverse algorithms has been introduced to enhance the efficiency and robustness of neural architecture generation. NAS algorithms fall into two main categories: One-Shot methods and black-box methods. A NAS method may not fall squarely into either category or may straddle both White et al. (2023). Black-box methods have been notable for their frequent use in the field. NAS strategies, including Bayesian optimization, evolutionary algorithms, and reinforcement learning, have been widely adopted Liu et al. (2021). However, one downside of these techniques is their significant computational cost, with some studies reporting the use of thousands of GPU days for their experiments Elsken et al. (2018); White et al. (2023). In contrast, one-shot methods have gained traction due to their considerable efficiency. These methods manage to generate promising results within a far shorter time span - typically a few GPU days, and in some cases, as reported by Dong et al. Dong & Yang (2019), even within a span of just a couple of hours.

The Differentiable Architecture Search (DARTS) paradigm, proposed by Liu et al. (2018), presents a novel approach to the automation of network architecture search Liu et al. (2018). DARTS is a unique combination of a cell-based search space and a gradient-based one-shot model, facilitating efficient exploration and evaluation of architectures. The search space in this context is realized as a Directed Acyclic Graph (DAG) where each edge of which can perform one out of 8 potential operations.

Auto-DeepLab (ADL) is a specialized variant of Differentiable Neural Architecture Search (NAS) that was developed to create effective architectures specifically for semantic segmentation tasks within the DeepLab framework Liu et al. (2019). Originating from the work of Liu et al., ADL enhances the DARTS-based, cell-centric search space Liu et al. (2018) by incorporating a hierarchical component to manage spatial resolution during the architecture search Elsken et al. (2022). In line with DeepLab conventions, the architecture search concludes with an Atrous Spatial Pyramid Pooling (ASPP) module Chen et al. (2017). However, unlike traditional DeepLab models, ADL utilizes only three branches in the ASPP module instead of the typical five Liu et al. (2019).

## 3 METHODS AND MODELS

### 3.1 CRITERIA AND METRICS

To study the effectiveness of transfer learning, we chose Auto-DeepLab (ADL) as our Neural Architecture Search (NAS) model. The selection was based on multiple criteria:

- **Computational Efficiency**: One-shot models like ADL significantly reduce the computational burden, making it feasible to perform multiple experiments.
- **Task Specificity**: ADL specializes in semantic segmentation, directly aligning with our research focus.
- **Documented Performance**: Previous works have validated ADL's effectiveness, providing a reliable starting point for our own evaluations White et al. (2023).

This NAS methodology serves as the underlying architecture for Experiment 1, where we intend to evaluate the role of transfer learning in semantic segmentation tasks.

### 3.2 TRAINING ENVIRONMENT AND DATA

Our experiments were conducted on a Computing Cluster equipped with NVIDIA A100 and V100 GPUs. Some tests also utilized an NVIDIA RTX 3090. The hardware environment is essential for Experiment 2, where we aim to understand the impact of dataset size on NAS outcomes. Furthermore, we contrast a model pre-trained on solar farm classification with one exclusively trained for segmentation tasks. We refer to these models as solis-transfer and solis-seg respectively.

A collection of over 200,000 Sentinel-2 level-2A images, serves as the empirical foundation of our research. Each image is a 224x224 pixel chip with 12 bands, and approximately half are positive examples featuring solar farms. Data size and quality are well-known to influence the performance and reliability of machine learning models. Experiment 2 will specifically delve into the impact of dataset size on the NAS process. To counter potential biases and overfitting, we employed a diverse

set of images from various geographical regions. Data augmentation techniques, including random horizontal and vertical flips, were applied to enhance model robustness.

### 3.3 Implementation and Parameter Selection

Our research utilized a PyTorch adaptation of the original AutoDeepLab model[2], optimized and modified for our dataset. This codebase serves as the foundation for all our experiments and is available for public scrutiny

In terms of parameter settings, we followed the guidelines set by Liu et al. Liu et al. (2019), with modifications to suit our specific hardware limitations. For instance, we adjusted the batch sizes to 22 or 12 depending on the available GPU memory. These parameter choices are especially relevant for Experiment 2, where we explore the NAS process under varying computational constraints.

### 3.4 Comparing NAS Results

#### 3.4.1 Objectives and Methodologies

Our research aims to provide a comprehensive evaluation of Auto-DeepLab's performance, particularly focusing on its adaptability to different input data sizes and types. This is directly tied to Experiment 2, which aims to understand how these factors influence the NAS process.

#### 3.4.2 Performance Evaluation of Final Models

Our final models will be trained on the complete Solis dataset, adhering to an 80/20 train-test split. This training regimen is aligned with our final experiment, where the best-performing model will be deployed in a real-world scenario to discover new solar farms.

- **Solis models**: The existing and new versions will serve as our primary benchmarks, particularly for Experiment 1 which focuses on transfer learning.
- **ADL-cs**: This model, found to be the best-performing by Liu et al. during their Cityscapes search Liu et al. (2019), will provide an external point of comparison.
- **ChatGPT generated model**: A randomly generated architecture will also be included as a lower-bound performance measure.

The primary metric is validation set mIoU, except for the Solis-transfer model where F1-score is used due to mIoU not having been captured during its training. The final test involves deploying the best model on untrained regions (the state of New York) to assess its generalization capabilities

## 4 Experimental Results

### 4.1 Experiment 1: Evaluating the Effectiveness of Transfer Learning

| Name | mIoU | F1-score |
|------|------|----------|
| Solis-seg | 0.9629 | 0.9621 |
| 10k-L | 0.9593 | 0.9582 |
| ADL-cs | 0.9586 | 0.9575 |
| 10k | 0.9567 | 0.9555 |
| ChatGPT | 0.9565 | 0.9552 |
| Solis-transfer | N/A | 0.89 |

Table 1: Top 5 models ranked by validation mIoU achieved during retraining.

The purpose of this experiment is to evaluate the effectiveness of transfer learning, particularly as employed by the Solis-transfer model. Our intention is to investigate if the prevalent approach of transfer learning from classification tasks remains the optimal strategy or if training directly on segmentation tasks from the outset can produce improved outcomes. To facilitate this analysis, we implemented a variant of the Solis model, Solis-seg, trained exclusively on segmentation.

Contrary to our expectations, not only did the Solis-seg model exhibit a marked performance

---

[2] https://github.com/NoamRosenberg/autodeeplab

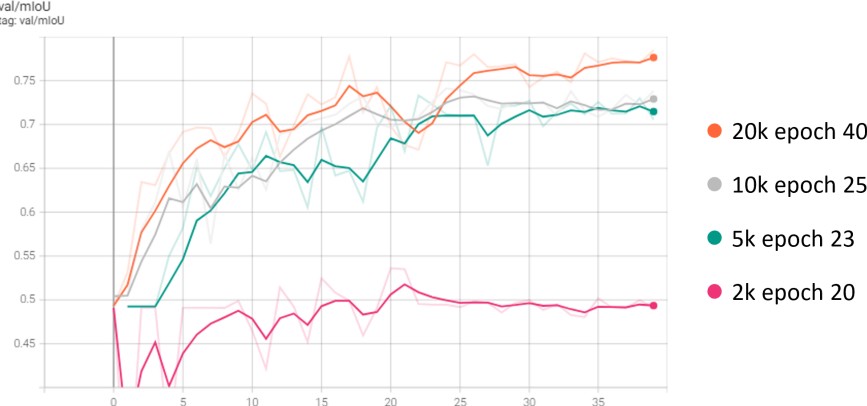

Figure 1: Validation mIoU for different dataset sizes during search.

improvement compared to Solis-transfer by increasing the best F1-score from 0.89 to 0.9621, it even ascended to the position of the highest-performing model. With an impressive final validation mIoU of 0.9629, it surpassed all the models obtained through our NAS experiments, emerging as the only model breaching the 0.96 threshold. Table 1 provides a summary of the top five models, ranked based on the mIoU scores achieved during the retraining phase. It underscores the dominance of Solis-seg in this experiment.

## 4.2 EXPERIMENT 2: ASSESSING THE IMPACT OF DATASET SIZE ON NAS

| Name | val mIoU (search) | val mIoU (retrain) | train mIoU (retrain) |
|------|-------------------|--------------------|----------------------|
| 10k | 0.741 | 0.9567 | 0.9653 |
| 2k | 0.536 | 0.9563 | 0.9637 |
| 5k | 0.733 | 0.9550 | 0.9630 |
| 20k | 0.785 | 0.9531 | 0.9607 |

Table 2: mIoU results for different dataset sizes.

In this experiment, we explored how the size of the dataset influences the outcome of Neural Architecture Search (NAS). Due to computational limitations, we opted for smaller subsets of the full dataset, specifically sizes of 2,000, 5,000, 10,000, and 20,000 images, referred to as 2k, 5k, 10k, and 20k. These subsets were considered to be representative samples for the purpose of architecture discovery.

During the search, we observe a correlation between the dataset size and the resulting validation mIoU as seen in Figure 1. The smallest dataset (2k) shows more variability in results, indicating sensitivity to data selection. Most of the searches reached peak performance shortly after 20 epochs, thus we scrutinize the structural components of the resulting architectures. Interestingly, despite similar performance metrics, the architectures exhibit considerable structural differences.

Upon retraining these architectures on the full dataset, the performance discrepancies observed during the search phase were not as pronounced. For example, the model trained on the largest dataset (20k) unexpectedly yielded the lowest performance when applied to the full dataset.

The results did not indicate a strong correlation between dataset size and final performance, suggesting that either an element of randomness was at play or that the subsets were sufficiently representative of the full dataset for this application.

## 4.3 DEPLOYING THE BEST MODEL TO FIND NEW SOLAR FARMS

In our final experiment, we deployed Solis-seg, our best-performing model, to detect new solar farms in satellite imagery covering New York State from 2022. The model identified 874 polygons, which, after accounting for multiple polygons representing single facilities, equate to approximately 583 potential solar farms. Figure 2 depicts a solar farm found by our model. Several of these locations are not documented in publicly available databases such as OpenStreetMap.

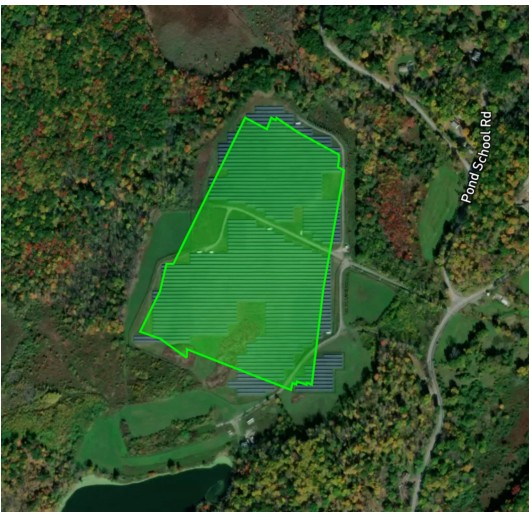

Figure 2: Example of a solar farm detected in New York state.

While Solis-seg was effective in identifying numerous solar farms, its performance was not as robust in the New York dataset as it was with the solar farms in our validation set. We noticed that the model detected some solar farms and entirely missed others, suggesting challenges in generalizing to new geographical regions (see Figure 3).

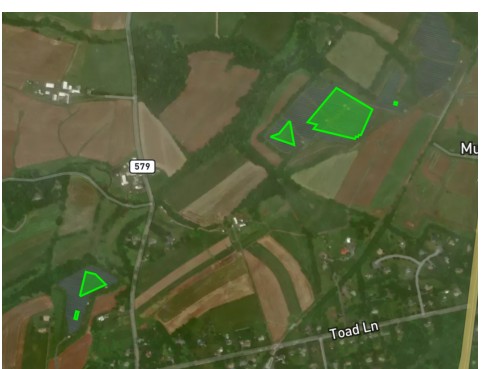

Figure 3: Example of a solar farm partially detected by the model.

This limitation underscores the importance of diverse training data, a notion supported by existing literature. The model's struggle to generalize indicates that it could benefit from a more diverse dataset that includes various architectural styles, landscapes, and environmental conditions.

Another challenge was the verification of the model's predictions due to the absence of up-to-date, high-resolution imagery. This issue made it difficult to determine whether certain polygons were indeed solar farms or false positives (see Figure 4).

Despite these challenges, Solis-seg's real-world deployment was largely successful. As a contribution to the community, we have made the dataset of detected solar farms in New York publicly available in a GitHub repository.[3] This addition enriches the growing body of solar farm data and can serve as a valuable resource for future research.

---

[3]To keep this submission anonymous, we do not include the URL here, and request that reviewers do not search for this data. If the paper is accepted to ICLR, we will include the URL in the camera-ready version of the paper.

# 5 DISCUSSION OF EXPERIMENTAL RESULTS

## 5.1 RE-EVALUATING THE EFFICACY OF TRANSFER LEARNING

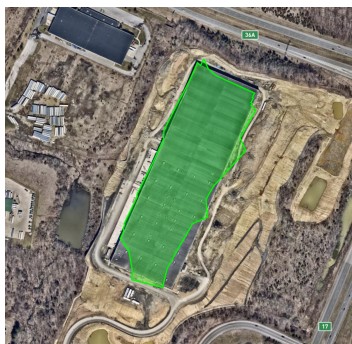

Figure 4: High-resolution image indicating a potential false positive.

The Solis-seg and Solis-transfer models differ solely in their training methodology as detailed in Section 4.1. Solis-seg is dedicated to the exclusive task of semantic segmentation of solar farms, whereas the ResNet component of Solis-transfer is initially trained to identify whether an image does or does not contain a solar farm (classification), and only thereafter it is trained for the task of segmentation.

Despite numerous trials with Solis-transfer, it has yet to surpass an F1 score of 0.89. In contrast, the single experiment conducted with Solis-seg yielded a significantly superior F1 score (0.962), clearly highlighting the effectiveness of task-specific training. The increase in performance is thus evidently attributable to the switch in training strategy, as no other alterations were made during the training process.

This surprising outcome suggests that the methods employed by the classification model to discern the presence of a solar farm differ considerably from the pixel-wise recognition performed during semantic segmentation. It posits the idea that the competencies required for these tasks might diverge to an extent that proficiency in one (classification) could potentially impede the ability to learn the other (segmentation).

Moreover, this experiment highlights the notion that the benefits of transfer learning are not universally applicable, but are contingent upon various factors including the degree of similarity between the source and target tasks, and the specific nature of these tasks. Our study, for example, points towards instances where a model specifically trained for a particular task from inception can outperform one that capitalizes on transferred knowledge from an ostensibly related task.

In summarizing our findings, it's compelling to note that our best-performing model surpassed the IoU score of 0.9 obtained by Kruitwagen et al. Kruitwagen et al. (2021). While an apples-to-apples comparison isn't feasible due to their employment of a considerably larger and globally distributed dataset, our results hold significance given the markedly higher relative score attained on our dataset.

## 5.2 THE ROBUSTNESS OF NAS IN SATELLITE IMAGE SEGMENTATION

Our research offers insights into applying Neural Architecture Search (NAS) for semantic segmentation of solar farms on Sentinel-2 imagery, the results of which were highlighted by our findings in Section 4.3. The uniformity of data quality across different dataset sizes and subsets resulted in little variation in performance among the various NAS-derived models. An exception to this was the model trained on the 20k dataset, which underperformed unexpectedly; see section 4.2. The precise reasons for this remain unclear, although disruptive data elements or an unfortunate random seed choice may be possible causes.

An intriguing finding from our experiments was that out of 14 NAS trials, only a single architecture managed to outperform any of the benchmarks, excluding Solis-transfer. This raises questions regarding the effectiveness and cost-benefit value of DARTS and Auto-DeepLab within this particular context, which will be further elaborated in Section 5.3.

Surprisingly, the randomly sampled architecture produced by ChatGPT outperformed almost all of the architectures identified via the architecture search. While this might be an outlier event and ad-

ditional random samples should be examined for validation, it raises questions about the consistency and effectiveness of the architecture search process in yielding superior architectures for certain use-cases even when it has demonstrated the capability to accurately assess comparative performance, as highlighted in Experiment 2.

Furthermore, it was observed that the performance of most models was closely aligned with that of the random model. This suggests that the search space may be densely populated with models that deliver comparable performance, thereby making it difficult to continually progress toward an optimal solution. This hypothesis is supported by studying the search graphs, particularly by the observation of most searches reaching their peak early. This pervasive challenge is credited by Chen and Hsieh Chen & Hsieh (2021) to DARTS' tendency to reach strong local minima in the search space.

Moreover, the top-performing NAS model, 10k-L, only slightly lagged behind the best-performing model, Solis-seg. This suggests that under appropriate conditions, NAS has the potential to generate architectures that approach or even match the state-of-the-art, even in specialized applications such as satellite imagery segmentation. The robustness and adaptability of NAS, despite the complexities and challenges, underscore its potential as a valuable technique.

### 5.3 COMPUTATIONAL TRADE-OFFS IN NAS APPLICATION

NAS is a demanding procedure, introducing substantial overhead to a machine learning pipeline. Not only does it necessitate training a model, it requires significant additional time and resources to discover the model architecture in the first place through search. This inherently prompts the question: When is the extra cost of performing NAS worthwhile?

| Dataset size | Search time (h) |
|---|---|
| 2k | 20 |
| 5k | 41 |
| 10k | 62 |
| 20k | 104 |

Table 3: Dataset size and search time.

In evaluating the efficiency of NAS, two main aspects come into play: the potential performance gain and the importance of this gain for the specific application. In our study, NAS proved to be less time-efficient when compared to traditional methods. Specifically, the Solis-seg model took 46 hours to train, while the average training time for NAS-derived architectures was around 59 hours. These figures do not yet account for the additional search time required by NAS, as shown in Table 3. When considering both the search and training times, the total computational time for NAS architectures vastly exceeds that for Solis-seg. This casts doubt on the cost-effectiveness of NAS, particularly when an off-the-shelf model like ResNet50-DeepLab performed best on our dataset after only 14 NAS trials.

Reflecting on the top five models derived from our study, as shown in Table 4, three out of the five top performers are baseline models that we originally proposed for comparison. Interestingly, even a randomly suggested model outperformed all but one model discovered through NAS.

While the search outcomes might not seem particularly outstanding — failing to surpass a ResNet-based model, marginally exceeding a model found by searching on a different dataset, and the curious case of a random model outperforming all but one NAS architecture — it is important to recognize that the top model found through the search, 10k-L, does not lag significantly behind the best model, Solis-seg.

There are potential improvements to our NAS process that could potentially enhance the performance of the discovered models. Though even if we were to conduct additional trials and come across a model that outperforms Solis-seg, the total cost of the new model would exceed the cost we incurred by training the off-the-shelf model by magnitude for the sake of a slight increase in performance.[4]

It's worth noting that all models outperformed Solis-transfer, implying that the DARTS search space is replete with viable model architectures. Additionally, given the low-resolution nature of the images in this study, this presents a relatively unconventional segmentation problem. Considering this,

---

[4]Scoring a perfect 1 should be impossible due to data imperfections.

the obtained results speak to the robustness and versatility of the models derived from the DARTS search space.

The decision of whether or not to use NAS essentially hinges on the importance of incremental performance improvement and the available alternatives to increase the performance of the model. In our case, however, it might be more productive to allocate resources toward enhancing other aspects of the model, such as augmenting the quality and volume of data Layman (2019) or investigating the optimal combination of spectral bands.

Moreover, the high computational cost of NAS could potentially deter smaller entities or individual researchers who operate with constrained computing resources. Without access to a computing cluster, this research project would have likely spanned well over a hundred continuous training days on an NVIDIA RTX-3090 GPU.

| Name | mIoU | F1-score |
|---|---|---|
| Solis-seg | 0.9629 | 0.9621 |
| 10k-L | 0.9593 | 0.9582 |
| ADL-cs | 0.9586 | 0.9575 |
| ChatGPT | 0.9565 | 0.9552 |
| 2k | 0.9563 | 0.9550 |
| Solis-transfer | N/A | 0.89 |

Table 4: The top 5 models ranked by validation mIoU obtained during retraining. The model 10k has been omitted here as it shares the same architecture as 10k-L. It would have been placed between ADL-cs and ChatGPT, see Table 1.

All these considerations should be factored in when deciding whether to employ NAS, further emphasizing the need for a case-by-case approach to the application of this technology.

Finally, it is also crucial to remember that NAS is a relatively nascent field. As with many emerging technologies, it will likely undergo considerable refinement and become more efficient and accessible in the coming years. Future advancements might mitigate many of the current limitations, enabling more widespread and accessible usage. As such, staying up to date on NAS development and its potential for evolving machine learning models will be critical to continuously evaluate its applications and benefits in the future.

## 6 CONCLUSION AND FUTURE WORK

Addressing the global need for renewable energy monitoring, this work introduces Solis-seg, a cutting-edge Deep Neural Network for solar farm detection in satellite imagery. With a record mean Intersection over Union (IoU) of 96.26% on a continental-scale dataset, the model sets new performance benchmarks. We demonstrate the practical application of Neural Architecture Search (NAS) in semantic segmentation, a largely unexplored domain for NAS. Our work shows that NAS methodologies can leverage additional image data, such as spectral bands, offering avenues for creating data-rich models in specialized tasks.

Contrary to popular practice, we question the efficacy of transfer learning from classification to semantic segmentation, suggesting that this approach may compromise performance. Our study also emphasizes the need to weigh the benefits of NAS against practical constraints like computational resources, particularly when computing resources are limited.

Future research endeavors could uncover further valuable insights by subjecting our model to the dataset employed by Kruitwagen et al. This approach would allow for the performance evaluation of our model in a more expansive and diverse setting. Unfortunately, developing a data pipeline, akin to the one employed by Kruitwagen et al., that synergizes their data with our trained model is likely a substantial undertaking due to the complex nature of these pipelines. This complexity is the primary reason we have not endeavored to attempt this in our current study.

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
