# OpenReview forum: "Detection and Segmentation of Solar Farms in Satellite Imagery: A Study of  Deep Neural Network Architectures"
_ICLR.cc/2024/Conference — Submitted to ICLR 2024_

### Official Review · Reviewer_rV62 · 2023-10-29

**Soundness:** 3 good
**Presentation:** 3 good
**Contribution:** 2 fair
**Rating:** 5
**Confidence:** 4

**Summary:**

This paper evaluates different network architectures - some pre-defined and
others found through neural architecture search (NAS) - on the task of solar
farm detection from satellite imagery.

The focus is on evaluation of Auto-DeepLab on this task. The NAS architectures
did not provide a benefit over the pre-defined models.

The authors consider a cost-benefit analysis of using NAS for these applied
tasks and conclude that the cost may often outweigh any potential benefit.

They report results of where this work was applied to detect undocumented solar
farms.

**Strengths:**

The result that it is sometimes difficult to transfer a classification model to
a segmentation task is useful (although similar results have been found and
exist in the literature - and those should be cited).

A random model from ChatGPT is interesting.

The work has applications to important real-world problems, and results from an initial deployment of this model are described.

The paper reports a negative result on the usefulness of NAS, which while not as exciting as positive results, is important to do.

**Weaknesses:**

* Vision transformers were not evaluated.

* Many of the details are unclear.

* The results seem to be reported on the validation set and there isn't a held out test set.

* Ultimately, the scope of this paper - while important - is limited. The experiments are a datapoint the applicability of NAS in the context of this task, but its difficult to draw any general conclusions. I think this work is more applicable to an application focused venue.

**Questions:**

> A collection of over 200,000 Sentinel-2 level-2A images

Is there a definition of what images are used? Is the dataset reproducible?

> This work introduces Solis-seg, a Deep Neural Network optimized

The abstract implies Solis-seg is a new architecture, but the details are never given. I'm not sure if Section 3.3 is implying that it is a variant of the original AutoDeepLab model? I'll assume it is, but this needs to be stated if you are saying you are introducing something.


> Our research utilized a PyTorch adaptation of the original AutoDeepLab model, optimized and modified for our dataset.

Details about what the important modifications were would be helpful.


> "after accounting for multiple polygons"

How were there multiple polygons on the same site? The details of polygon
generation is unclear? Binary or another form of thresholding from a heatmap?


> equate to approximately 583 potential solar farms

Potential? Is there a process for validating the proposals? It would be helpful to report how many of these were validated as actual correct decisions. I.e. in a random sample, have a group classify them as false positives, true positives, and indeterminate cases and report the numbers.


> ChatGPT generated model

The details of how this worked or what type of model it produced is unclear.
Did it just produce pytorch code?


Have you seen "An Analysis of Pre-Training on Object Detection" (https://arxiv.org/pdf/1904.05871.pdf) which concludes that its hard to tune a detection network from a classification network, but easy to go the other way? This seems similar to your experience of having trouble with classification and segmentation.


The total time to train models was given, but how many batch iterations / second was each model able to perform at? This number can help compare model efficiency directly and abstracts away "unfortunate random seeds" that might cause one model to take longer to train than another.


All scores in Table 4 are reasonably high. Is there a chance they are overtuned to the validation data?

At the very end you mention you use all spectral bands? Did you evaluate using RGB only? or RGB+NIR? In my experience the coastal, swir, and other bands do not perform as well as RGB or RGBN on sentinel 2 tasks. I'm not sure why this is, but it's an effect I've noticed.


Did you measure the power consumption and emissions from your work? If not, I
recommend codecarbon.

---

> ### Author Response · Authors · 2023-11-23
> **Thanks for the review, please check the updated paper**
>
> We appreciate the review and in particular its appreciation of the importance of this work.  Indeed, the focus of the paper is on  the application of known techniques but in a new setting, thus producing new knowledge.  Our belief is that the generation of renewable energy is important enough to warrant the publication of such application-centric papers at ICLR.   Many of the reviwer's requests for further details have been considered when creating the updated paper.

---

### Official Review · Reviewer_vhXk · 2023-10-29

**Soundness:** 1 poor
**Presentation:** 2 fair
**Contribution:** 1 poor
**Rating:** 1
**Confidence:** 4

**Summary:**

The work presents a study over different deep neural network architecture for solar panel segmentation. The study includes known approaches i.e., deep neural network architecture and architectures discovered by neural architecture search. Additionally one architecture is derived from ChatGPT. Architectures are benchmarked in two experiments (a) against each other, and (b) against the size of the training dataset.

**Strengths:**

**S1**: the paper aims at a very important domain, which is centered around the netzero target as outlined in the Paris Agreement. With respect to this, the paper focuses on clean energy production and within this area it aims at solar panel detection. I personally think that machine learning in combination with earth observation can significantly contribute to quantifying globally important key performance indicators on either the consumption or the production side of the netzero targets.

**Weaknesses:**

**W1**: the submitted paper does only provide little to no methodological contribution or methodological novelty. All methods presented in the paper are known and applied to the given dataset. Specifically, for the NAS based architecture benchmark, it seems that only a few selected architectures are evaluated or benchmarked.

**W2**: Important details to understand the method and experimental setup are missing:
- (i) no details are given for the architectures to be evaluated. What are the results of NAS? Are these architectures based on U-Net or are they FCN?
- (ii) no information is given about the classification task, which was used for the proposed transfer learning experiment. Just at the end of the paper, there is a sentence about a ResNet being used (page 7). How was the transfer learning experiment setup? Did the authors take the pretrained ResNet model parameter as the encoder for the U-Net and randomly initialized the decoder?
- (iii) what was the prompt used to retrieve the ChatGPT architecture?
- (iv) what is the "solis-dataset" used for training and evaluation (Sec. 3.4.2)? Is this the author's own dataset of 200k Sentinel-2 level-2A images? Where is the solar farm segmentation ground truth coming from? Is it coming from one of the related works? If yes, why not take the entire dataset from the related work for evaluation?

Such details are important and do provide valuable information about the approach helping to understand the author's contribution. I would welcome to see such details in the appendix of the paper.

**W3**: the authors call the ChatGPT model a "random model". This is confusing. What do the authors mean by calling the model random? Do they mean the architecture or the model parameter being random.  I am asking because the experiments in the paper do also focus on the evaluation of pre-trained models+transfer learning and models trained from scratch. In case the architecture is meant to be "random", I would not agree to call this "random". ChatGPT retrieved the architecture from somewhere. It would be interesting if the provided architecture from ChatGPT is a known (or a modification of a) segmentation architecture such as U-Net or is something totally different? The authors might even try to ask ChatGPT to provide the sources of the architecture. Since all such details are missing, I am only speculating what this "random model" is about.

**W4**: structure of the paper could improve. The experimental setup is fragmented in multiple sections so that it is difficult to follow. It can be found in "Sec. 3.1 Criteria and Metrics" and "Sec. 3.2 Training Environment and Data" while "Sec 3.1" is mixing up computational resources with dataset.


**W5**: image captions do not really convey what the figure is about. I would suggest providing more details in the image caption. Additionally, some figures do not provide enough details to be understood since labels for the x or y-axis are missing.

**W6**: It is not clear what Table 2 is about (what is search, what is retrain?) and is not referenced in the text body of the paper. Since it probably conveys results of one of the two experiments of the submission it might be important to put the table in context of the main body of the paper.

**Questions:**

Please see weaknesses.

---

> ### Author Response · Authors · 2023-11-23
> **Thanks for the review, please check the updated paper**
>
> We appreciate the review and in particular its appreciation of the importance of this work.  Indeed, the focus of the paper is in the application of known techniques but in a new setting, thus producing new knowledge.  Particular points of note in the updated paper include: clarification of its focus and contributions, added information about ChatGPT (see appendix) as well as other reproducibility information, improvements to the paper's structure.

---

### Official Review · Reviewer_2B7A · 2023-11-01

**Soundness:** 1 poor
**Presentation:** 1 poor
**Contribution:** 2 fair
**Rating:** 1
**Confidence:** 4

**Summary:**

The paper focuses on identifying solar farms from satellite images. The work uses an existing NAS approach for semantic segmentation to segment solar farms in Sentinel 2 images .

**Strengths:**

1) The idea of detecting solar farm from remote sensing images is interesting and linked with SDGs.

**Weaknesses:**

1) The presentation needs improvement. The paper is oddly structured with an incoherent sections. Each section appears like a standalone text with little continuity. The paper presents the motivation (detection of solar farms) and immediately discusses the challenges of NAS. Is NAS the ONLY approach to detect solar farm? If the author wants to use NAS for solar farm detection, it should be clearly specified beforehand. Also, other approaches for solar farm detection should also be presented and compared with NAS. Also, wouldn't it be better to summarize the major contribution of the work before discussing the related work.

2) This work appears as an implementation of an existing method (AutoDeepLab) for solan farm identification. However, it is very difficult to confirm this because the paper does NOT have a method section. The section 3, which ought to be presenting the methods,  starts with metrics and training strategies!

Due to a lack of clarity in the novelty offered by this work, along with a very poor presentation, I am inclined to reject this work.

**Questions:**

Please refer to my comments in the weakness section.

---

### Official Review · Reviewer_Co8H · 2023-11-01

**Soundness:** 2 fair
**Presentation:** 2 fair
**Contribution:** 1 poor
**Rating:** 3
**Confidence:** 3

**Summary:**

In this paper, the authors present a study on the use of Neural Architecture Search (NAS) for semantic segmentation of solar farms in satellite imagery. The study consists of two types of experiments. The first experiment examines both transfer learning and non-transfer learning-based semantic segmentation, while the second experiment investigates the impact of dataset size on the performance of the NAS. The authors achieved an impressive 96.26% (mIoU) on the European dataset.

**Strengths:**

1- In this work, the authors investigate the use of NAS (Neural Architecture Search) for identifying solar farms from satellite images, which is a new and innovative application of NAS.
2- Additionally, instead of limiting themselves to just NAS, the authors have also implemented a transfer learning-based approach for semantic segmentation and have compared the outcomes of both methods.

**Weaknesses:**

1- The work presented here doesn’t offer new ideas and has a very limited contribution.
2- There are no ablation studies included to explain the reasoning behind the performance.
3- Although this is an application-based study, there aren’t enough case studies provided to demonstrate the applicability of the implemented NAS architecture. It would have been better if more experiments with additional datasets were included.

**Questions:**

1.     It is recommended to conduct further experiments on various datasets to investigate the potential of NAS in the analysis of satellite imagery.
2- Additionally, providing ablation studies could help explain the rationale behind the performance of the NAS architecture.
3- Also, it is unclear how a model is randomly selected from ChatGPT for the lower bound. Therefore, more information should be provided on the selection process of this model.

---

> ### Author Response · Authors · 2023-11-23
> **Thanks for the review, please check the updated paper**
>
> We appreciate the review and have updated the paper accordingly.   In particular, we have clarified the focus and contributions of the paper and added information about ChatGPT (see appendix) as well as other reproducibility information.  However, the inclusion of other similar datasets has not been feasible, due to the lack of availability, and our belief that the current study in itself is important and stands on its own.

---

### Meta-Review · Area_Chair_QBm8 · 2023-12-04

**Metareview:**

Dear authors,

Thank you for submitting the draft. Unfortunately, majority of the authors have indicated that this draft is not ready for the publication. We encourage authors to take inconsideration reviewers' comments when updating the draft.

regards

Meta-reviewer

**Justification For Why Not Higher Score:**

Majority of reviews were not favorable.

**Justification For Why Not Lower Score:**

N/A

---

### Decision · Program_Chairs · 2024-01-16

Reject